# Prognostic value of the Residual Cancer Burden index according to breast cancer subtype: Validation on a cohort of BC patients treated by neoadjuvant chemotherapy

**Anne-Sophie Hamy[1,2], Lauren Darrigues[3], Enora Laas[3], Diane De Croze[4], Lucian Topciu[5], Giang-Thanh Lam[3,6], Clemence Evrevin[2], Sonia Rozette[2], Lucie Laot[3], Florence Lerebours[2], Jean-Yves Pierga[2,3], Marie Osdoit[3], Matthieu Faron[7], Jean-Guillaume Feron[3], Marick Laé[5], Fabien Reyal[2,3]***

**1** Residual Tumor & Response to Treatment Laboratory, RT2Lab, INSERM, U932 Immunity and Cancer, Institut Curie, Paris, France, **2** Department of Medical Oncology, Institut Curie, Saint-Cloud, France, **3** Department of Surgery, Faculté de Médecine Descartes, Université Paris, Institut Curie, Paris, France, **4** Department of Tumor Biology, Institut Curie, Saint-Cloud, France, **5** Department of Tumor Biology, Institut Curie, Paris, France, **6** Department of Gynecology and Obstetrics, Geneva University Hospitals, Geneva, Switzerland, **7** Department of biostatistics and epidemiology, INSERM 1018 CESP Oncostat team, Gustave Roussy Cancer Campus, Villejuif, France

* fabien.reyal@curie.fr

**Data Availability Statement:** All relevant data are within the paper and its Supporting Information files.

## Abstract

### Introduction

The Residual Cancer Burden (RCB) quantifies residual disease after neoadjuvant chemotherapy (NAC). Its predictive value has not been validated on large cohorts with long-term follow up. The objective of this work is to independently evaluate the prognostic value of the RCB index depending on BC subtypes (Luminal, *HER2*-positive and triple negative (TNBCs)).

### Methods

We retrospectively evaluated the RCB index on surgical specimens from a cohort of T1-T3NxM0 BC patients treated with NAC between 2002 and 2012. We analyzed the association between RCB index and relapse-free survival (RFS), overall survival (OS) among the global population, after stratification by BC subtypes.

### Results

717 patients were included (luminal BC (n = 222, 31%), TNBC (n = 319, 44.5%), *HER2*-positive (n = 176, 24.5%)). After a median follow-up of 99.9 months, RCB index was significantly associated with RFS. The RCB-0 patients displayed similar prognosis when compared to the RCB-I group, while patients from the RCB-II and RCB-III classes were at increased risk of relapse (RCB-II *versus* RCB-0: HR = 3.25 CI [2.1–5.1] *p*<0.001; RCB-III *versus* RCB-0: HR = 5.6 CI [3.5–8.9] *p*<0.001). The prognostic impact of RCB index was significant for TNBC and *HER2*-positive cancers; but not for luminal cancers ($P_{\text{interaction}}$ = 0.07). The

**Funding:** We thank Roche France for financial support for the construction of the Institut Curie neoadjuvant database (NEOREP). Funding was also obtained from the Site de Recherche Integrée en Cancérologie/Institut National du Cancer (Grant No. INCa-DGOS-4654). A-S Hamy-Petit was supported by an ITMO-INSERM-AVIESAN translational cancer research grant. The funders had no role in study design, data collection and analysis, decision to publish, or preparation of the manuscript.

**Competing interests:** We thank Roche France for financial support for the construction of the Institut Curie neoadjuvant database (NEOREP). Funding was also obtained from the Site de Recherche Integrée en Cancérologie/Institut National du Cancer (Grant No. INCa-DGOS-4654). A-S Hamy-Petit was supported by an ITMO-INSERM-AVIESAN translational cancer research grant.The authors have declared that no competing interests of any kind exist, with funders such as Roche France, or with other entities. This does not alter our adherence to PLOS ONE policies on sharing data and materials.

prognosis of RCB-III patients was poor (8-years RFS: 52.7%, 95% CI [44.8–62.0]) particularly in the TNBC subgroup, where the median RFS was 12.7 months.

## Conclusion

RCB index is a reliable prognostic score. RCB accurately identifies patients at a high risk of recurrence (RCB-III) with TNBC or *HER2*-positive BC who must be offered second-line adjuvant therapies.

## Introduction

**Neoadjuvant chemotherapy (NAC),** *i.e.* chemotherapy before surgery, is currently being used in poor prognosis breast carcinoma. Besides reducing tumor burden and enabling conservative breast surgery, NAC also provides a unique opportunity to evaluate the response of patients with breast cancer (BC) to different treatments. Pathologic complete response (pCR), defined as an absence of invasive cancer in the breast and axillary lymph nodes, is associated with better long-term survival, though its prognostic value is particularly important in highly aggressive tumors, such as triple negative breast cancer (TNBC) and *HER2*-positive BC [1].

However, since most of the tumors do not achieve pCR following NAC, this binary indicator gathers in a unique category the majority of the patients, thus precluding the opportunity to sharply predict their oncological outcome. While patients with pCR exhibit an excellent prognosis, a wide clinical heterogeneity remains within those patients failing to reach complete response, and the identification of patients with residual disease at a high risk of relapse is a substantial challenge. Hence, the subdivision of the BC population into several prognostic groups could help improving the prediction of survival benefits [2].

The Residual Cancer Burden (RCB) index has been developed in 2007 by Symmans and colleagues from the M.D. Anderson Cancer Center (MDACC) [3] to quantify residual disease following NAC. The RCB index combines pathological findings in the primary tumor bed and the regional lymph nodes to calculate a continuous index. This index is subdivided into four classes with an increasing amount of residual disease: RCB 0 (pCR), RCB-I, RCB-II, and RCB-III. Several prospective clinical studies (CALGB [4], GEICAM [5] and I-SPY [6]) included RCB as a primary or secondary end point for response to NAC. Among the variety of scoring systems developed in the last years (CPS [7], CPS+EG, Neobioscore ([8]), RCB index was recommended by the BIG-NABCG (Breast International Group-North American Breast Cancer Group) to quantify residual disease in neoadjuvant trials, in addition to pCR [9]. However, so far, it remains unknown if RCB index displays high prognostic performances in routine practice, and external fully independent validation of the prognostic value of the RCB index is lacking. The current study aims at evaluating the prognostic value of the RCB index in a large real-life cohort of breast cancer patients treated with NAC.

## Material and methods

### Patients and tumors

The analysis was performed on a previously described cohort of patients [10] with invasive breast carcinoma stage T1-T3NxM0 and treated with NAC at Institut Curie, Paris, between 2002 and 2012 (NEOREP Cohort, CNIL declaration number 1547270). We included unilateral, non-recurrent, non-inflammatory, non-metastatic tumors, excluding T4 tumors. All

patients received NAC, followed by surgery and radiotherapy. NAC regimens changed over our recruitment period (anthracycline-based regimen or sequential anthracycline-taxanes regimen), with trastuzumab used in an adjuvant and/or neoadjuvant setting since 2005. Endocrine therapy (tamoxifen or aromatase inhibitor) was prescribed when indicated. The study was approved by the Breast Cancer Study Group of Institut Curie and was conducted according to institutional and ethical rules regarding research on tissue specimens and patients. Written informed consent from the patients was not required by French regulations.

## Tumor samples

Cases were considered estrogen receptor (ER) or progesterone receptor (PR) positive (+) if at least 10% of the tumor cells expressed estrogen and/or progesterone receptors (ER/PR), in accordance with guidelines used in France [11]. *HER2* expression was determined by immunohistochemistry with scoring in accordance with American Society of Clinical Oncology (ASCO)/College of American Pathologists (CAP) guidelines [12]. Scores 3+ were reported as positive, score 1+/0 as negative (-). Tumors with scores 2+ were further tested by FISH. *HER2* gene amplification was defined in accordance with ASCO/CAP guidelines [12]. We evaluated a mean of 40 tumor cells per sample and the mean *HER2* signals per nuclei was calculated. A *HER2*/CEN17 ratio ≥ 2 was considered positive, and a ratio < 2 negative [12]. BC subtypes were defined as follows: tumors positive for either ER or PR, and negative for *HER2* were classified as luminal; tumors positive for *HER2* were considered to be *HER2*-positive BC; tumors negative for ER, PR, and *HER2* were considered to be triple-negative breast cancers (TNBC). Tumor cellularity was defined as the percentage of tumor cells (in situ and invasive) on the specimen (biopsy or surgical specimen). Mitotic index was reported per 10 high power fields (HPF) (1 HPF = 0.301 mm2).

## Pathological review

717 pathological pretreatment core needle biopsies and the corresponding post-NAC surgical specimens were reviewed independently by two experts in breast diseases (ML, DD).

**Residual Cancer Burden index.** Histological components of the "Residual Cancer Burden" were retrieved for calculating the score as described in 2007 by Symmans (see S1 File). RCB index enables the classification of residual disease into four categories: RCB-0 (complete pathologic response = pCR), RCB-I (minimal residual disease), RCB-II (moderate residual disease) and RCB-III (extensive residual disease). Two cutoff points defined those subgroups: the first (RCB-III *v* RCB-I/II) was selected as the 87th percentile (RCB, 3.28), and the second (RCB-I *v* RCB-II) corresponded to the 40th percentile (RCB, 1.36). RCB has been calculated through the web-based calculator that is freely available on the internet (www.mdanderson.org/breastcancer_RCB).

**TILs and LVI.** Lymphovascular invasion (LVI) was defined as the presence of carcinoma cells within a finite endothelial-lined space (a lymphatic or blood vessel). Tumor infiltrating lymphocytes (TILs) were defined as the presence of mononuclear cells infiltrate (including lymphocytes and plasma cells, excluding polymorphonuclear leukocytes), and were also evaluated retrospectively, for research purposes (see S1 File).

## Study endpoints

Relapse-free survival (RFS) was defined as the time from surgery to death, loco-regional recurrence or distant recurrence, whichever occurred first. Overall survival (OS) was defined as the time from surgery to death. The date of last known contact was retained for patients for whom none of these events were recorded. Survival cutoff date analysis was February, 1st, 2019.

## Statistical analysis

The study population was described in terms of frequencies for qualitative variables, or medians and associated ranges for quantitative variables. Chi-square tests were performed to search for differences between subgroups for each variable (considered significant for p-values $\leq$ 0.05). Survival probabilities were estimated by the Kaplan–Meier method, and survival curves were compared in log-rank tests. Hazard ratios and their 95% confidence intervals were calculated with the Cox proportional hazards model. Variables with a p-value for the likelihood ratio test equal to 0.05 or lower in univariate analysis were selected for inclusion in the multivariate analysis. A forward stepwise selection procedure was used to establish the final multivariate model and the significance threshold was 5%. Missing data were present in 69 out of 717 patients (9.6%) for the variable lymphovascular invasion and we imputed these missing data by a chained equation multiple imputation strategy, as recommended in a previous study [13]. We assessed the effect of the RCB index both on the hazard scale (with a proportional hazards model) and on the log-of-time scale with an accelerated failure time model. Data were processed and statistical analyses were carried out with R software version 3.1.2 (www.cran.r-project.org, [14].

## Results

### Patients' characteristics

A total of 717 patients were included in the cohort. Patients and tumors characteristics are summarized in Table 1. Median age was 48 years old (range [25–80]) and most of the patients (63%) were premenopausal. BC repartition by subtype was as follows: luminal (n = 222; 31%), TNBC (n = 319; 44.5%), *HER2*-positive (n = 176; 24.5%). No difference was observed according to BC subtype regarding age, menopausal status, clinical tumor size nor clinical nodal status. TNBC and *HER2*-positive BCs were associated with a higher grade, Ki67 and mitotic index than luminal BCs ($p<0.001$).

### RCB index repartition and patients' characteristics by RCB class

At NAC completion, RCB-0 (pCR) was observed in 202 patients (28.2%). Among 515 patients with residual disease, RCB index repartition was as follows RCB-I: n = 65 (9%), RCB-II: n = 309 (43.1%) and RCB-III: n = 141 (19.7%) (Table 2, Fig 1A). The RCB index distribution was significantly different by BC subtypes (p<0.001) (Table 2, Fig 1B): luminal tumors were more likely to be classified as RCB-II (49.1%) or III (36.9%), whereas TNBC or *HER2*-positive BC were more likely to be RCB-0 or I (45.7% and 52.3% respectively) ($p<0.001$). Only small subsets of TNBC and *HER2*-positive BCs were classified as RCB-III (13.2% and 8.5% respectively).

The distribution of the index was bimodal as previously described [3], and the 2 modes of the distribution strongly overlapped with the post-NAC nodal status (Fig 2A and 2B). Most of the patients with tumors classified as RCB-I were free from axillar nodal involvement, while the majority of patients with tumors classified as RCB-III had a node-positive disease (Fig 2C and 2D).

Patients' characteristics by RCB class are summarized in Table 2 and are represented in Fig 3. Among pre-NAC parameters, RCB class was significantly different by tumor size (p<0.001) (Fig 3A), clinical nodal status ($p<0.001$) (Fig 3B), pathological grade ($p<0.001$) (Fig 3C), BMI ($p<0.05$) (Fig 3D) and mitotic index ($p<0.001$) (Fig 3E). Pre-NAC TILs were inversely associated with RCB ($p<0.001$) (Fig 3F). Among the post-NAC parameters, higher RCB class was significantly correlated with the presence of LVI ($p<0.001$; Fig 3G), nodal involvement

**Table 1. Patients' characteristics among the whole population and in each subtype.**

| Characteristics | Class | All | Luminal | TNBC | HER2 | p |
|---|---|---|---|---|---|---|
| n = | | 717 (100%) | 222 (31.0%) | 319 (44.5%) | 176 (24.5%) | |
| Pre-NAC characteristics | | | | | | |
| Age (years) | <45 | 285 (39.7%) | 88 (39.6%) | 119 (37.3%) | 78 (44.3%) | 0.531 |
| | 45–55 | 254 (35.4%) | 82 (36.9%) | 118 (37.0%) | 54 (30.7%) | |
| | >55 | 178 (24.8%) | 52 (23.4%) | 82 (25.7%) | 44 (25.0%) | |
| Menopausal | pre | 451 (63.5%) | 146 (66.4%) | 191 (60.6%) | 114 (65.1%) | 0.350 |
| status | post | 259 (36.5%) | 74 (33.6%) | 124 (39.4%) | 61 (34.9%) | |
| | [19–25] | 414 (57.8%) | 121 (54.5%) | 176 (55.3%) | 117 (66.5%) | **0.046** |
| BMI | <19 | 41 (5.7%) | 18 (8.1%) | 16 (5.0%) | 7 (4.0%) | |
| | >25 | 261 (36.5%) | 83 (37.4%) | 126 (39.6%) | 52 (29.5 | |
| Clinical tumor size | T1 | 47 (6.6%) | 10 (4.5%) | 27 (8.5%) | 10 (5.7%) | 0.199 |
| | T2 | 481 (67.1%) | 160 (72.1%) | 207 (64.9%) | 114 (64.8%) | |
| | T3 | 189 (26.4%) | 52 (23.4%) | 85 (26.6%) | 52 (29.5%) | |
| Clinical | N0 | 282 (39.4%) | 79 (35.7%) | 141 (44.2%) | 62 (35.2%) | 0.061 |
| nodal status | N1-N2-N3 | 434 (60.6%) | 142 (64.3%) | 178 (55.8%) | 114 (64.8%) | |
| Histology | NST | 660 (92.6%) | 199 (89.6%) | 291 (92.1%) | 170 (97.1%) | **0.017** |
| | others | 53 (7.4%) | 23 (10.4%) | 25 (7.9%) | 5 (2.9%) | |
| Grade | I-II | 211 (30.1%) | 119 (55.1%) | 40 (12.8%) | 52 (30.1%) | **<0.001** |
| | III | 490 (69.9%) | 97 (44.9%) | 272 (87.2%) | 121 (69.9%) | |
| Ki67 | <20 | 33 (18.4%) | 8 (50.0%) | 22 (15.5%) | 3 (14.3%) | **0.003** |
| | ≥20 | 146 (81.6%) | 8 (50.0%) | 120 (84.5%) | 18 (85.7%) | |
| Mitotic Index | ≤22 | 389 (57.0%) | 153 (72.9%) | 124 (41.2%) | 112 (65.5%) | **<0.001** |
| | >22 | 293 (43.0%) | 57 (27.1%) | 177 (58.8%) | 59 (34.5%) | |
| ER status | negative | 396 (55.2%) | 0 (0.0%) | 319 (100.0%) | 77 (43.8%) | **<0.001** |
| | positive | 321 (44.8%) | 222 (100.0%) | 0 (0.0%) | 99 (56.2%) | |
| PR status | negative | 473 (68.2%) | 43 (21.1%) | 319 (100.0%) | 111 (64.9%) | **<0.001** |
| | positive | 221 (31.8%) | 161 (78.9%) | 0 (0.0%) | 60 (35.1%) | |
| HER2 status | negative | 541 (75.5%) | 222 (100.0%) | 319 (100.0%) | 0 (0.0%) | **<0.001** |
| | positive | 176 (24.5%) | 0 (0.0%) | 0 (0.0%) | 176 (100.0%) | |
| NAC | AC | 61 (8.5%) | 3 (1.4%) | 54 (16.9%) | 4 (2.3%) | **<0.001** |
| regimen | AC-Taxanes | 576 (80.3%) | 202 (91.0%) | 222 (69.6%) | 152 (86.4%) | |
| | others | 80 (11.2%) | 17 (7.7%) | 43 (13.5%) | 20 (11.4%) | |
| Post-NAC characteristics | | | | | | |
| RCB | RCB-0 | 202 (28.2%) | 11 (5.0%) | 123 (38.6%) | 68 (38.6%) | **<0.001** |
| | RCB-I | 65 (9.1%) | 18 (8.1%) | 23 (7.2%) | 24 (13.6%) | |
| | RCB-II | 309 (43.1%) | 109 (49.1%) | 131 (41.1%) | 69 (39.2%) | |
| | RCB-III | 141 (19.7%) | 84 (37.8%) | 42 (13.2%) | 15 (8.5%) | |
| Number of | 0 | 445 (62.1%) | 78 (35.1%) | 238 (74.6%) | 129 (73.3%) | **<0.001** |
| nodes involved | 1–3 | 188 (26.2%) | 100 (45.0%) | 49 (15.4%) | 39 (22.2%) | |
| | ≥4 | 84 (11.7%) | 44 (19.8%) | 32 (10.0%) | 8 (4.5%) | |
| LVI | no | 500 (77.2%) | 130 (66.0%) | 232 (80.8%) | 138 (84.1%) | **<0.001** |
| | yes | 148 (22.8%) | 67 (34.0%) | 55 (19.2%) | 26 (15.9%) | |

Missing values: menopausal status n = 7; BMI n = 1; clinical nodal status n = 1; mitotic index n = 35; histology n = 4; grade n = 16; Ki67 n = 538; LVI n = 69.

Abbreviations: pCR = pathological complete response; BMI = body mass index; NST = no special type; ER = oestrogen receptor; PR = progesterone receptor; NAC = neoadjuvant chemotherapy; AC = anthracyclines; LVI = lymphovascular invasion; RCB = residual cancer burden.

**Table 2. Patients' characteristics according to RCB classes.**

| Variable | Class | pCR | RCB-I | RCB-II | RCB-III | p |
|---|---|---|---|---|---|---|
| n = | | 202 (28.2%) | 65 (9.1%) | 309 (43.1%) | 141 (19.7%) | |
| Pre-NAC parameters | | | | | | |
| Age (years) | <45 | 76 (37.6%) | 31 (47.7%) | 130 (42.1%) | 48 (34.0%) | 0.136 |
| | 45–55 | 66 (32.7%) | 25 (38.5%) | 108 (35.0%) | 55 (39.0%) | |
| | >55 | 60 (29.7%) | 9 (13.8%) | 71 (23.0%) | 38 (27.0%) | |
| Menopausal | pre | 119 (59.8%) | 46 (71.9%) | 202 (65.6%) | 84 (60.4%) | 0.235 |
| Status | post | 80 (40.2%) | 18 (28.1%) | 106 (34.4%) | 55 (39.6%) | |
| BMI | 19≤BMI≤25 | 125 (62.2%) | 46 (70.8%) | 176 (57.0%) | 67 (47.5%) | **0.007** |
| | <19 | 8 (4.0%) | 6 (9.2%) | 15 (4.9%) | 12 (8.5%) | |
| | >25 | 68 (33.8%) | 13 (20.0%) | 118 (38.2%) | 62 (44.0%) | |
| Tumoral Size | T1 | 26 (12.9%) | 3 (4.6%) | 12 (3.9%) | 6 (4.3%) | **<0.001** |
| | T2 | 129 (63.9%) | 52 (80.0%) | 213 (68.9%) | 87 (61.7%) | |
| | T3 | 47 (23.3%) | 10 (15.4%) | 84 (27.2%) | 48 (34.0%) | |
| Nodal status | N0 | 83 (41.1%) | 32 (49.2%) | 138 (44.7%) | 29 (20.7%) | **<0.001** |
| pre NAC | N1-N2-N3 | 119 (58.9%) | 33 (50.8%) | 171 (55.3%) | 111 (79.3%) | |
| Mitotic Index | ≤22 | 89 (47.1%) | 40 (66.7%) | 167 (56.2%) | 93 (68.4%) | **0.001** |
| | >22 | 100 (52.9%) | 20 (33.3%) | 130 (43.8%) | 43 (31.6%) | |
| Histology | NST | 188 (93.5%) | 59 (90.8%) | 292 (95.4%) | 121 (85.8%) | **0.004** |
| | other | 13 (6.5%) | 6 (9.2%) | 14 (4.6%) | 20 (14.2%) | |
| Grade | I-II | 33 (16.8%) | 21 (32.8%) | 91 (30.2%) | 66 (47.5%) | **<0.001** |
| | III | 164 (83.2%) | 43 (67.2%) | 210 (69.8%) | 73 (52.5%) | |
| Ki67 | <20% | 6 (10.2%) | 3 (30.0%) | 17 (20.7%) | 7 (25.0%) | 0.198 |
| | ≥20% | 53 (89.8%) | 7 (70.0%) | 65 (79.3%) | 21 (75.0%) | |
| TILs stromal | mean % | 34 | 26.1 | 19.7 | 19.0 | **<0.001** |
| Subtype | luminal | 11 (5.4%) | 18 (27.7%) | 109 (35.3%) | 84 (59.6%) | **<0.001** |
| | TNBC | 123 (60.9%) | 23 (35.4%) | 131 (42.4%) | 42 (29.8%) | |
| | HER2 | 68 (33.7%) | 24 (36.9%) | 69 (22.3%) | 15 (10.6%) | |
| ER status | negative | 163 (80.7%) | 31 (47.7%) | 152 (49.2%) | 50 (35.5%) | **<0.001** |
| | positive | 39 (19.3%) | 34 (52.3%) | 157 (50.8%) | 91 (64.5%) | |
| PR status | negative | 183 (91.5%) | 38 (60.3%) | 185 (61.3%) | 67 (51.9%) | **<0.001** |
| | positive | 17 (8.5%) | 25 (39.7%) | 117 (38.7%) | 62 (48.1%) | |
| HER2 status | negative | 134 (66.3%) | 41 (63.1%) | 240 (77.7%) | 126 (89.4%) | **<0.001** |
| | positive | 68 (33.7%) | 24 (36.9%) | 69 (22.3%) | 15 (10.6%) | |
| NAC Regimen | AC | 17 (8.4%) | 3 (4.6%) | 30 (9.7%) | 11 (7.8%) | 0.599 |
| | AC-Taxanes | 158 (78.2%) | 57 (87.7%) | 244 (79.0%) | 117 (83.0%) | |
| | others | 27 (13.4%) | 5 (7.7%) | 35 (11.3%) | 13 (9.2%) | |
| Post-NAC parameters | | | | | | |
| Nodal involvment | 0 | 202 (100.0%) | 53 (81.5%) | 188 (60.8%) | 2 (1.4%) | **<0.001** |
| | 1–3 | 0 (0.0%) | 12 (18.5%) | 101 (32.7%) | 75 (53.2%) | |
| | ≥4 | 0 (0.0%) | 0 (0.0%) | 20 (6.5%) | 64 (45.4%) | |
| LVI | no | 200 (99.0%) | 41 (91.1%) | 190 (71.4%) | 69 (51.1%) | **<0.001** |
| | yes | 2 (1.0%) | 4 (8.9%) | 76 (28.6%) | 66 (48.9%) | |
| Mitotic Index | mean, SD | | 0.82 (2.54) | 17.75 (28.88) | 19.32 (33.53) | **<0.001** |
| TILs stromal | mean, SD | 8.7 (10.5) | 12.8 (14.6) | 14.8 (12.5) | 15.2 (14) | **<0.001** |

Abbreviations: pCR = pathological complete response; BMI = body mass index; NST = no special type; ER = oestrogen receptor; PR = progesterone receptor; NAC = neoadjuvant chemotherapy; AC = anthracyclines; LVI = lymphovascular invasion, TILs = tumor infiltrating lymphocytes.

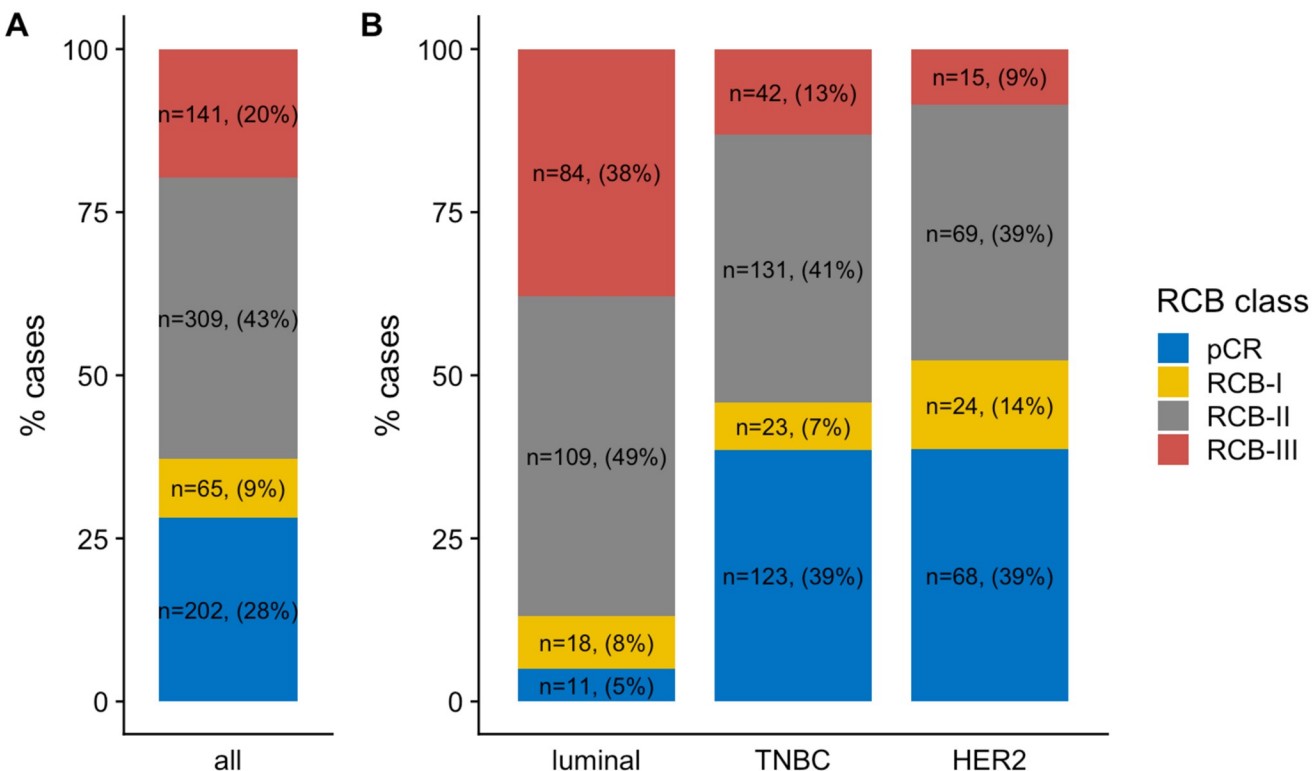

**Fig 1. RCB class distribution among the whole population and by BC subtypes: A) among the whole population, B) in each subtype population.**

($p < 0.001$; Fig 3H), whereas post-NAC TILs were positively associated with RCB ($p < 0.001$; Fig 3I).

## Survival analyses

With a median follow-up of 99.9 months, [range (9.3–184.2 months)], 205 patients experienced relapse, and 133 were deceased. Among the whole population, RCB was significantly associated with RFS (Fig 4A), and this association was significant after both univariate analysis and multivariate analysis (Table 3). Eight-years relapse free survival was good in RCB-0 and RCB-I group (89.9%, CI [85.6–94.4] and 89.0% CI [80.7–98.2] respectively), whereas the prognosis was intermediate in RCB-II patients (67.8%, CI [62.4–73.5]) and poor in RCB-III patients (52.7%, CI [44.8–62.0]). Increasing RCB was associated with an increased risk of relapse (RCB-II versus RCB-0: HR = 3.25 CI [2.1–5.1] $p < 0.001$ and RCB-III versus RCB-0: HR = 5.6 CI [3.5–9.0] $p < 0.001$). The prognosis impact of the RCB index was significant in TNBC and *HER2*-positive BCs, but not in luminal BC (Fig 4B–4D and S1-S3 Tables in S1 File) ($P_{interaction}$ = 0.05, though the interaction test failed to reach statistical significance after multivariate analysis ($P_{interaction}$ = 0.057)). In addition to the increased risk of relapse, RCB index was also significantly associated with an earlier time-to-relapse, as estimated by the AFT regression model (RCB II *versus* RCB 0 and I grouped, HR = 3.27, 95% CI [2.18–4.91], RCB-III *versus* RCB 0 and I grouped, HR = 5.73, 95% CI [3.74–8.76] $p < 0.001$), and this was true in TNBC and *HER2*-positive BCs ($p < 0.001$) but not in luminal BCs ($p = 0.06$). In TNBC, RCB-III class identified a group of patients with extremely poor prognosis, displaying a median relapse-free survival of 12.7 months. We also identified an interaction between post-NAC TILs and RCB class to predict RFS ($P_{interaction}$ = 0.03). Post-NAC TILs had no prognostic impact on

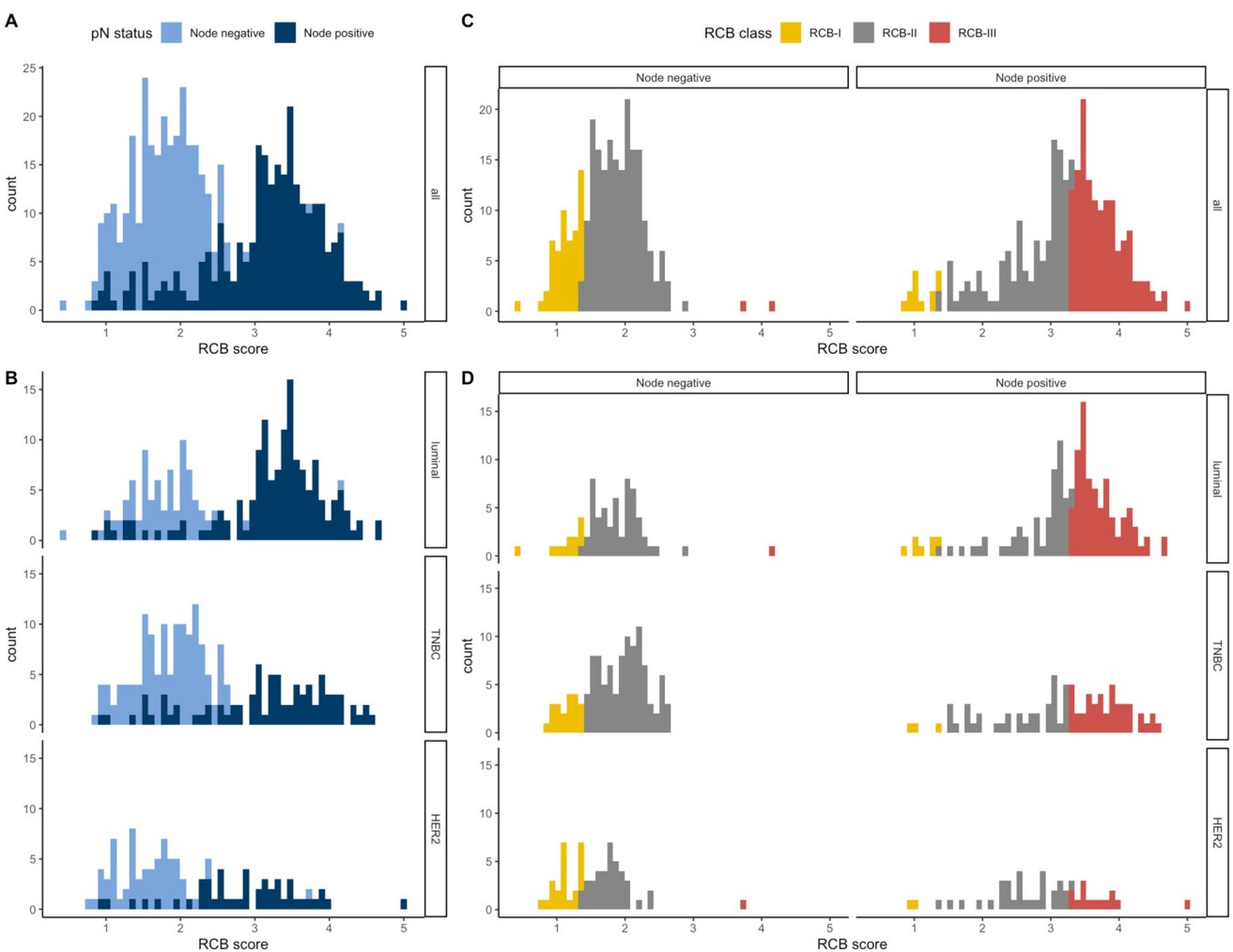

**Fig 2.** Histograms of the distribution of residual cancer burden (RCB) index in the patients with residual disease at surgery immediately following NAC, in the whole population (A) and according to phenotype of disease (B). Histograms showing the distribution of nodal status (positive or negative) according to the RCB value, in the whole population (C) and according to the phenotype of disease (D).

RFS in pCR, RCB-I and RCB-II subgroups, while post-NAC TILs had a poor prognostic impact (HR = 1.019, [1.001–1.037]) in the RCB-III subgroup.

Overall survival analyses yielded similar results (Fig 5, Table 4, S4-S6 Tables in S1 File). Together with BC subtype, RCB index was the only independent predictor of survival in the whole population.

## Discussion

In this retrospective reanalysis of 717 surgical specimens of BC patients treated with NAC with a long-term follow-up, we confirm the strong prognostic value of the RCB index.

RCB index was first created in 2007 by Symmans and colleagues on a cohort of 241 BC patients who completed NAC [3]. In this study, patients had almost a two-fold increase in relapse risk for each unit of increase in the RCB index and it remained significantly associated with the risk of disease recurrence after multivariate analysis. Though RCB is a composite end-point built upon 6 variables, this index was shown to be highly reproducible. Peintinger *et al.*

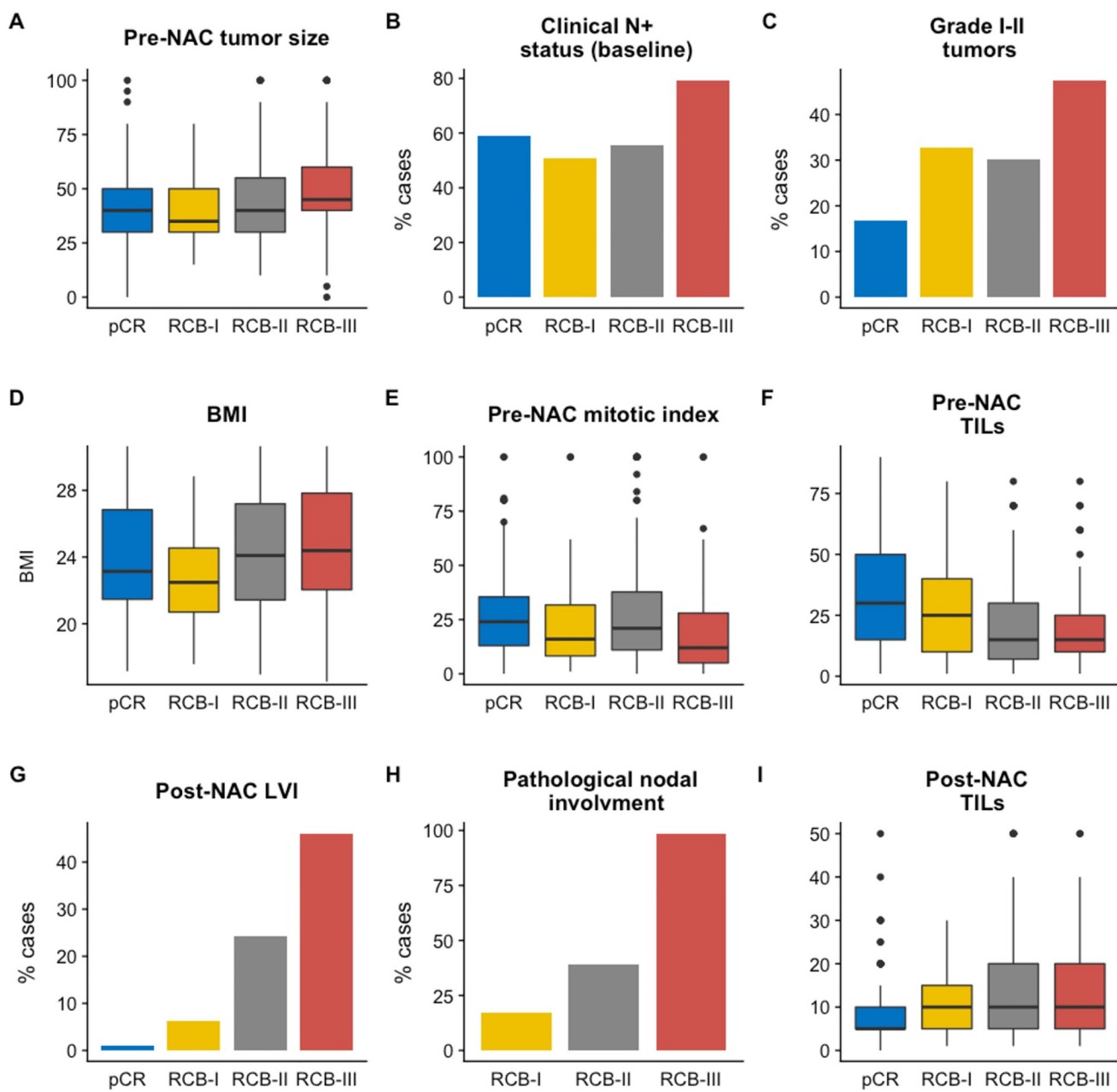

**Fig 3. Associations between RCB classes (pCR to RCB-III) and clinico-pathological variables: A) pre-NAC tumor size, B) Clinical Nodal status at baseline, C) Grade I to II tumors, D) BMI, E) pre-NAC mitotic index, F) Pre-NAC TILs, G) post-NAC LVI, H) pathological nodal involvement, I) post-NAC TILs.**

retrospectively assessed RCB on a series of 100 pathology slides from BC cases treated by NAC, and the overall concordance was 0.93 (95%CI = 0.91–0.95) after an independent review by five pathologists [15]. However, so far, the prognostic value of the index was evaluated only in small studies ([3, 16–22] (S7 Table in S1 File). To the best of our knowledge, we report here the largest fully independent cohort available with a long-term follow-up, with a notably high number of patients with TNBCs.

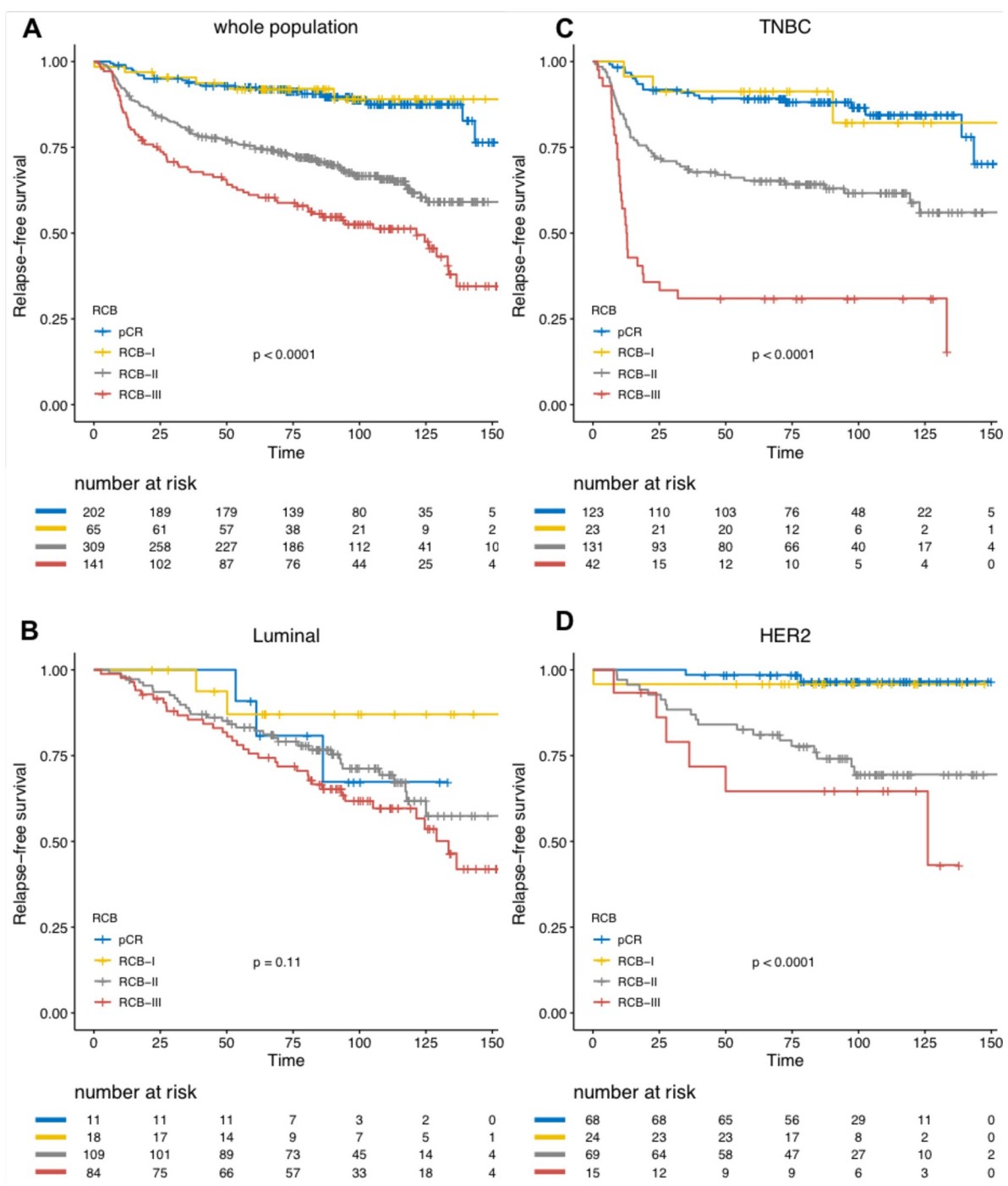

**Fig 4. Association of RCB classes (pCR to III) with relapse-free survival (RFS): A) whole population (N = 717), B) luminal tumors (N = 222), C) TNBC (N = 319), D) HER2-positive BC (N = 176).**

Several findings of our study are of interest. *First*, in line with the findings of Symmans and colleagues, we found that the prognosis of patients with RCB-I was not significantly different than the prognosis of patients whose tumor reached pCR (RCB-0). The latter finding confirms that the category of pCR patients, known to be at a very low-risk of relapse, could be extended

**Table 3. Association of clinical and pathological pre and post-NAC parameters with relapse-free survival after univariate and multivariate analysis in the whole population.**

| Variable | Class | Number | Events | Univariate | | | | Multivariate | | |
|---|---|---|---|---|---|---|---|---|---|---|
| | | | | HR | CI | p* | p | HR | CI | p |
| Pre-NAC parameters | | | | | | | | | | |
| Age (years) | <45 | 285 | 92 | 1 | | | 0.266 | | | |
| | 45–55 | 254 | 67 | 0.81 | [0.59–1.11] | | | | | |
| | >55 | 178 | 46 | 0.78 | [0.55–1.11] | | | | | |
| Menopausal status | pre | 451 | 127 | 1 | | | 0.796 | | | |
| | post | 259 | 74 | 1.04 | [0.78–1.38] | | | | | |
| BMI | 19≤BMI≤25 | 414 | 111 | 1 | | | 0.348 | | | |
| | <19 | 41 | 11 | 1.01 | [0.54–1.87] | | | | | |
| | >25 | 261 | 83 | 1.23 | [0.93–1.64] | | | | | |
| Tumor size | T1 | 47 | 13 | 1 | | | **0.027** | | | |
| | T2 | 481 | 127 | 0.93 | [0.53–1.65] | 0.812 | | | | |
| | T3 | 189 | 65 | 1.41 | [0.77–2.55] | 0.263 | | | | |
| Clinical nodal status | N0 | 282 | 78 | 1 | | | 0.597 | | | |
| | N1-N2-N3 | 434 | 127 | 1.08 | [0.81–1.43] | | | | | |
| Mitotic index | ≤22 | 389 | 110 | 1 | | | 0.185 | | | |
| | >22 | 293 | 90 | 1.21 | [0.91–1.6] | | | | | |
| Histology | NST | 660 | 182 | 1 | | | **0.026** | | | |
| | other | 53 | 22 | 1.65 | [1.06–2.57] | | | | | |
| Grade | I-II | 211 | 70 | 1 | | | 0.268 | | | |
| | III | 490 | 131 | 0.85 | [0.63–1.13] | | | | | |
| Ki67 | <20% | 33 | 10 | 1 | | | 0.53 | | | |
| | ≥20% | 146 | 51 | 1.24 | [0.63–2.45] | | | | | |
| Subtype | luminal | 222 | 75 | 1 | | | <0.001 | 1 | - | - |
| | TNBC | 319 | 102 | 1.1 | [0.82–1.49] | 0.523 | | 2,13 | [1.54–2.96] | <0.001 |
| | HER2 | 176 | 28 | 0.46 | [0.3–0.71] | <0.001 | | 0,92 | [0.58–1.45] | 0,7 |
| ER status | negative | 396 | 112 | 1 | | | 0.516 | | | |
| | positive | 321 | 93 | 0.91 | [0.69–1.2] | | | | | |
| PR status | negative | 473 | 135 | 1 | | | 0.26 | | | |
| | positive | 221 | 59 | 0.84 | [0.62–1.14] | | | | | |
| HER2 status | negative | 541 | 177 | 1 | | | <0.001 | | | |
| | positive | 176 | 28 | 0.43 | [0.29–0.65] | | | | | |
| NAC regimen | AC | 61 | 25 | 1 | | | 0.115 | | | |
| | AC-Taxanes | 576 | 161 | 0.66 | [0.43–1] | | | | | |
| | Others | 80 | 19 | 0.58 | [0.32–1.06] | | | | | |
| TILs | (continuous) | | | 0,99 | [0.98–0.99] | | **0,002** | | | |
| Post-NAC parameters | | | | | | | | | | |
| Nodal involvment | 0 | 445 | 86 | 1 | | | <0.001 | | | |
| | 1–3 | 188 | 69 | 2 | [1.45–2.74] | <0.001 | | | | |
| | ≥4 | 84 | 50 | 3.85 | [2.71–5.45] | <0.001 | | | | |
| RCB class | pCR | 202 | 23 | 1 | | | <0.001 | 1 | - | - |
| | RCB-I | 65 | 7 | 0.98 | [0.42–2.3] | 0.972 | | 1,17 | [0.50–2.74] | 0.48 |
| | RCB-II | 309 | 102 | 3.25 | [2.07–5.11] | <0.001 | | 3,38 | [2.11–5.39] | <0.001 |
| | RCB-III | 141 | 73 | 5.61 | [3.51–8.97] | <0.001 | | 6,29 | [3.73–10.62] | <0.001 |
| Interaction term RCB class*BC subtype | | | | | | | 0,051 | | | |
| Interaction term RCB class*Post-NAC TILs | | | | | | | 0,058 | | | |

(*Continued*)

**Table 3.** (Continued)

| Variable | Class | Number | Events | Univariate | | | | Multivariate | | |
|---|---|---|---|---|---|---|---|---|---|---|
| | | | | HR | CI | p* | p | HR | CI | p |
| LVI | no | 500 | 108 | 1 | | | <0.001 | 1 | - | - |
| | yes | 148 | 75 | 2.76 | [2.06–3.71] | <0.001 | | 1,55 | [1.15–2.08] | 0,004 |
| TILs | (continuous) | | | 1,01 | [0.99–1.02] | | 0,311 | | | |

Abbreviations: pCR = pathological complete response; BMI = body mass index; NST = no special type; ER = oestrogen receptor; PR = progesterone receptor; NAC = neoadjuvant chemotherapy; AC = anthracyclines; TILs = tumor infiltrating lymphocytes; RCB = residual cancer burden; LVI = lymphovascular invasion.

to patients with minimal residual disease. **Second**, we also confirm the very poor prognosis of patients with RCB-III disease, particularly in TNBC patients where the post-NAC median RFS barely exceeded one year. The identification of poor-prognosis after NAC is of substantial importance, as data from the CREATE-X and the KATHERINE trials suggest that these patients may benefit from the addition of adjuvant capecitabine [23] in the TNBC subpopulation, or adjuvant TDM-1 in *HER2*-positive BCs respectively [24]. In the latter trials, both second-line therapies were associated with a decrease of the recurrence risk, nearly reaching 50%. **Finally**, patients with RCB-II disease displayed an intermediate prognosis, and it remains unknown if they would benefit from additional therapies. As they represent 40% of the cohort, further prognostic subsetting using genomic signatures or additional clinical or pathological features should be of particular interest in this group.

In our cohort, RCB index displayed a strong discriminative power in TNBC and *HER2*-positive BC but not in luminal BCs, and we identified a trend towards an interaction ($P_{interaction}$ = 0.07) between BC subtype and RCB class. However, a pooled meta-analysis of more than 5000 individual RCB data with long-term follow up was recently presented by Symmans and colleagues [25]. In this study, RCB was significantly associated with BC outcomes, even in the luminal BC subgroup. These results are consistent with a lack of power to detect such differences in our data, where the subgroup of patients with luminal subtype who achieved pCR or RCB-1 only included 29 patients, therefore leading to a low number of events. This finding is also consistent with the well-known fact that BC subtypes respond differentially to NAC [26], and that the prognostic value of pCR is greatest in aggressive tumor subtypes such as TNBC or *HER2*-positive BC [1, 27] than in luminal BCs. Of note, Symmans and colleagues previously published the SET index signature assaying 165 genes from ER-related transcription. On a cohort of 131 patients with ER+ BC treated with prior neoadjuvant chemotherapy, both the RCB index and the SET index were independently predictive of the distant relapse risk and the elevated endocrine sensitivity was associated with reduced relapse risk when there was less than extensive RCB after chemotherapy [28]. In this context, the validation of the SET index signature in an independent NAC-treated cohort would be of interest.

Last, our study opens new perspectives for further improvement of the RCB index. We recently demonstrated that the presence of lymphovascular invasion (LVI) after NAC was associated with a dramatically impaired relapse-free survival in a BC subtype-dependent manner [29], and we show here that this feature adds an independent prognostic information to the RCB in the whole population, and in every BC subtype but luminal BCs. We also previously pointed out an interaction between RCB and the presence of stromal immune infiltration after chemotherapy [30], and identified an impaired prognostic impact of post-NAC TILs in the RCB-III subgroup. As immunotherapy is increasingly becoming part of the therapeutic strategy of breast cancer [31–35], the combination of both patterns could be an efficient tool to select poor-prognosis patients likely to benefit from such innovative treatments [9].

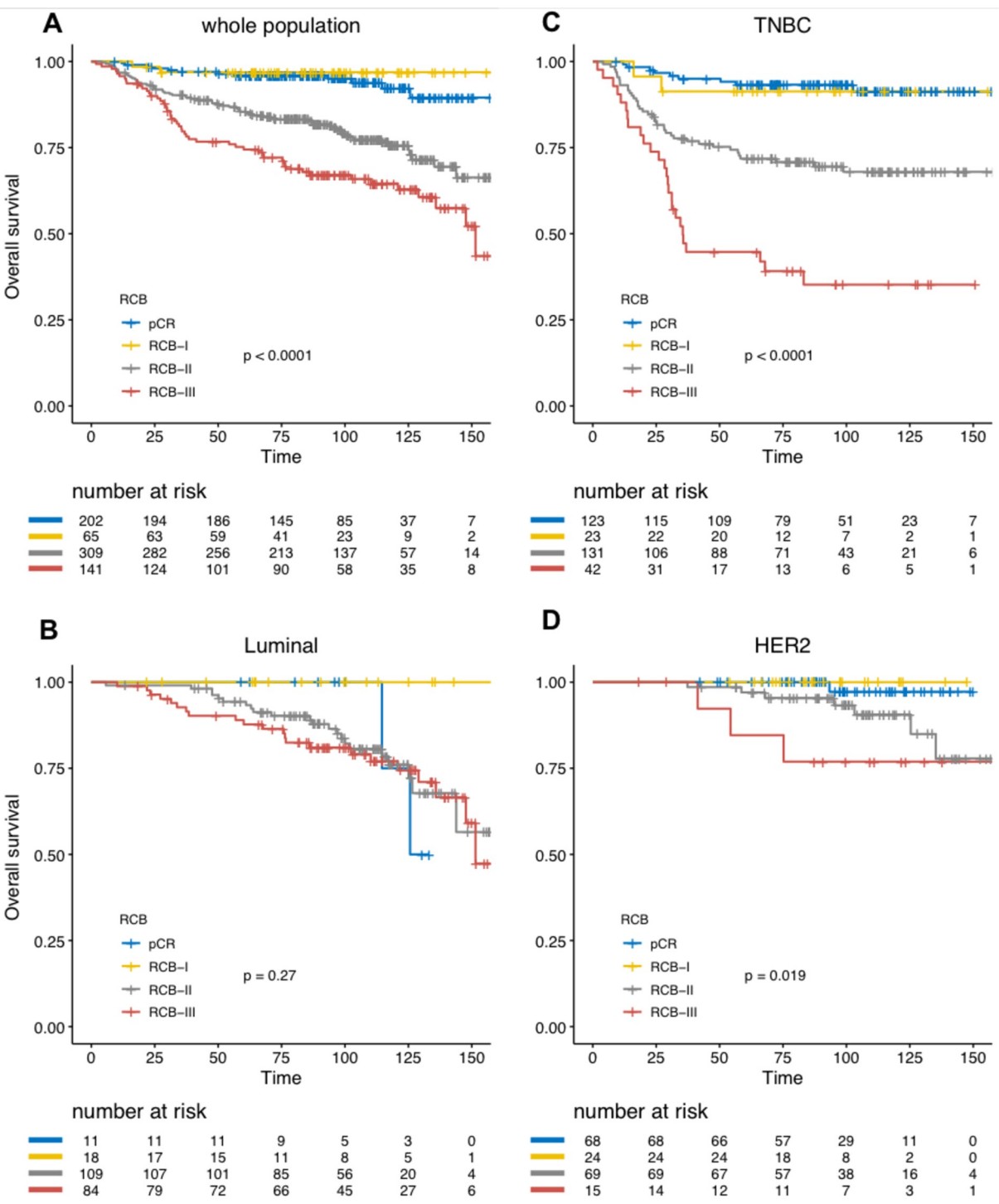

**Fig 5. Association of RCB classes (0 to III) with overall survival (OS): A) whole population (N = 717), B) luminal tumors (N = 222), C) TNBC (N = 319), D) HER2-positive BC (N = 176).**

**Table 4. Association of clinical and pathological pre and post-NAC parameters with overall survival after univariate and multivariate analysis in the whole population.**

| Variable | Class | Number | Events | Univariate | | | | Multivariate | | |
|---|---|---|---|---|---|---|---|---|---|---|
| | | | | HR | CI | $p^*$ | $p$ | HR | CI | $p$ |
| Pre-NAC parameters | | | | | | | | | | |
| **Age (years)** | <45 | 285 | 57 | 1 | | | 0.514 | | | |
| | 45–55 | 254 | 48 | 0.96 | [0.65–1.41] | | | | | |
| | >55 | 178 | 28 | 0.77 | [0.49–1.21] | | | | | |
| **Menopausal status** | pre | 451 | 80 | 1 | | | 0.457 | | | |
| | post | 259 | 51 | 1.14 | [0.8–1.62] | | | | | |
| **BMI** | 19≤BMI≤25 | 414 | 74 | 1 | | | 0.837 | | | |
| | <19 | 41 | 8 | 1.14 | [0.55–2.36] | | | | | |
| | >25 | 261 | 51 | 1.1 | [0.77–1.58] | | | | | |
| **Tumor size** | T1 | 47 | 8 | 1 | | | **0.007** | 1 | - | - |
| | T2 | 481 | 78 | 0.92 | [0.44–1.91] | 0.823 | | 0.74 | [0.35–1.55] | 0.422 |
| | T3 | 189 | 47 | 1.64 | [0.77–3.47] | 0.197 | | 1.23 | [0.57–2.66] | 0.594 |
| **Clinical nodal status** | N0 | 282 | 48 | 1 | | | 0.463 | | | |
| | N1-N2-N3 | 434 | 85 | 1.14 | [0.8–1.63] | | | | | |
| **Mitotic index** | ≤22 | 389 | 64 | 1 | | | 0.014 | | | |
| | >22 | 293 | 66 | 1.54 | [1.09–2.18] | **0.014** | | | | |
| **Histology** | ductal | 660 | 121 | 1 | | | 0.65 | | | |
| | other | 53 | 11 | 1.15 | [0.62–2.14] | | | | | |
| **Grade** | I-II | 211 | 36 | 1 | | | 0.291 | | | |
| | III | 490 | 94 | 1.23 | [0.84–1.81] | | | | | |
| **Ki67** | <20 | 33 | 5 | 1 | | | 0.33 | | | |
| | ≥20 | 146 | 35 | 1.59 | [0.62–4.07] | | | | | |
| **Subtype** | luminal | 224 | 19 | 1 | | | **<0.001** | 1 | - | - |
| | TNBC | 311 | 59 | 2.77 | [1.65–4.65] | 0.075 | | 2.7 | [1.8–4.05] | **<0.001** |
| | HER2 | 181 | 3 | 0.24 | [0.07–0.83] | **<0.001** | | 0.51 | [0.24–1.08] | 0.078 |
| **ER status** | negative | 396 | 80 | 1 | | | **0.049** | | | |
| | positive | 321 | 53 | 0.71 | [0.5–1] | **0.049** | | | | |
| **PR status** | negative | 473 | 93 | 1 | | | 0.052 | | | |
| | positive | 221 | 33 | 0.67 | [0.45–1] | 0.052 | | | | |
| **HER2 status** | negative | 541 | 122 | 1 | | | **<0.001** | | | |
| | positive | 176 | 11 | 0.25 | [0.13–0.46] | **<0.001** | | | | |
| **NAC regimen** | AC | 61 | 13 | 1 | | | 0.489 | | | |
| | AC-Taxanes | 576 | 110 | 0.96 | [0.54–1.72] | | | | | |
| | Others | 80 | 10 | 0.65 | [0.29–1.49] | | | | | |
| **TILs** | (continuous) | | | 0,99 | [0.98–0.99] | | **0,01** | | | |
| Post-NAC parameters | | | | | | | | | | |
| **Nodal involvment** | 0 | 445 | 51 | 1 | | | **<0.001** | | | |
| | 1–3 | 188 | 46 | 2.1 | [1.41–3.13] | **<0.001** | | | | |
| | ≥4 | 84 | 36 | 4.24 | [2.76–6.5] | **<0.001** | | | | |
| **RCB class** | pCR | 202 | 12 | 1 | | | **<0.001** | 1 | - | - |
| | RCB-I | 65 | 2 | 0.55 | [0.12–2.45] | 0.43 | | 0.75 | [0.17–3.38] | 0.711 |
| | RCB-II | 309 | 68 | 3.85 | [2.09–7.12] | **<0.001** | | 4.17 | [2.21–7.86] | **<0.001** |
| | others | 141 | 51 | 6.59 | [3.51–12.37] | **<0.001** | | 6.6 | [3.28–13.27] | **<0.001** |
| **LVI** | no | 500 | 66 | 1 | | | **<0.001** | 1 | - | - |
| | yes | 148 | 55 | 3.07 | [2.15–4.39] | **<0.001** | | 1.76 | [1.21–2.57] | **0.003** |

(*Continued*)

**Table 4.** (Continued)

| Variable | Class | Number | Events | Univariate | | | | Multivariate | | |
|---|---|---|---|---|---|---|---|---|---|---|
| | | | | HR | CI | $p^*$ | $p$ | HR | CI | p |
| TILs | (continuous) | | | 0,99 | [0.99–1.02] | | 0,329 | | | |

Abbreviations: pCR = pathological complete response; BMI = body mass index; NST = no special type; ER = oestrogen receptor; PR = progesterone receptor; NAC = neoadjuvant chemotherapy; AC = anthracyclines; TILs = tumor infiltrating lymphocytes; RCB = residual cancer burden; LVI = lymphovascular invasion.

## Supporting information

**S1 File.**
(PDF)

## Author Contributions

**Conceptualization:** Anne-Sophie Hamy, Enora Laas, Fabien Reyal.

**Data curation:** Anne-Sophie Hamy, Lauren Darrigues, Enora Laas, Clemence Evrevin, Sonia Rozette.

**Formal analysis:** Anne-Sophie Hamy, Lauren Darrigues, Enora Laas, Diane De Croze, Lucian Topciu, Giang-Thanh Lam, Florence Lerebours, Jean-Yves Pierga, Marie Osdoit, Matthieu Faron, Jean-Guillaume Feron, Marick Laé, Fabien Reyal.

**Funding acquisition:** Anne-Sophie Hamy.

**Investigation:** Anne-Sophie Hamy.

**Methodology:** Anne-Sophie Hamy, Enora Laas, Fabien Reyal.

**Project administration:** Anne-Sophie Hamy, Fabien Reyal.

**Resources:** Anne-Sophie Hamy.

**Software:** Anne-Sophie Hamy, Enora Laas.

**Supervision:** Anne-Sophie Hamy, Enora Laas, Fabien Reyal.

**Validation:** Anne-Sophie Hamy, Enora Laas, Fabien Reyal.

**Visualization:** Anne-Sophie Hamy.

**Writing – original draft:** Anne-Sophie Hamy, Lauren Darrigues.

**Writing – review & editing:** Anne-Sophie Hamy, Lucie Laot, Fabien Reyal.

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
