## [Decision Letter · Decision Letter 0]

18 Feb 2020

PONE-D-19-31211

Prognostic value of the Residual Cancer Burden index according to breast cancer subtype: validation on a cohort of BC patients treated by neoadjuvant chemotherapy.

PLOS ONE

Dear Dr Reyal,

Thank you for submitting your manuscript to PLOS ONE. After careful consideration, we feel that it has merit but does not fully meet PLOS ONE’s publication criteria as it currently stands. Therefore, we invite you to submit a revised version of the manuscript that addresses the points raised during the review process.

We would appreciate receiving your revised manuscript by Apr 03 2020 11:59PM. To enhance the reproducibility of your results, we recommend that if applicable you deposit your laboratory protocols in protocols.io, where a protocol can be assigned its own identifier (DOI) such that it can be cited independently in the future. For instructions see: http://journals.plos.org/plosone/s/submission-guidelines#loc-laboratory-protocols

We look forward to receiving your revised manuscript.

Kind regards,

Tina Hieken, MD

Academic Editor

PLOS ONE

Journal Requirements:

2.  We noticed you have some minor occurrence(s) of overlapping text with the following previous publication(s), which needs to be addressed:

https://doi.org/10.1093/annonc/mdx309

In your revision ensure you cite all your sources (including your own works), and quote or rephrase any duplicated text outside the Methods section. Further consideration is dependent on these concerns being addressed.

3. In the ethics statement in the manuscript and in the online submission form, please provide additional information about the patient records/samples used in your retrospective study. Specifically, please ensure that you have discussed whether all data/samples were fully anonymized before you accessed them and/or whether the IRB or ethics committee waived the requirement for informed consent. If patients provided informed written consent to have data/samples from their medical records used in research, please include this information.

4. Please include your tables as part of your main manuscript and remove the individual files. Please note that supplementary tables (should remain/ be uploaded) as separate "supporting information" files

Additional Editor Comments (if provided):

Reviewers' comments:

Reviewer's Responses to Questions

**Comments to the Author**

1. Is the manuscript technically sound, and do the data support the conclusions?

Reviewer #1: Yes

Reviewer #2: Yes

Reviewer #3: Yes

2. Has the statistical analysis been performed appropriately and rigorously? 

Reviewer #1: Yes

Reviewer #2: Yes

Reviewer #3: Yes

3. Have the authors made all data underlying the findings in their manuscript fully available?

Reviewer #1: Yes

Reviewer #2: Yes

Reviewer #3: Yes

4. Is the manuscript presented in an intelligible fashion and written in standard English?

Reviewer #1: Yes

Reviewer #2: Yes

Reviewer #3: Yes

5. Review Comments to the Author

Reviewer #1: This is a well-written manuscript with a worthy study goal. My only comment would be that the TILS data seemed a bit out of place in this study. I would focus this solely on the validation of the RCB index. Data related to LVI and TILS association with RCB is probably saved for a separate manuscript.

Reviewer #2: This article is an independent validation study of the RCB index within the main subtypes of breast cancer and with long-term follow-up. Furthermore, it addresses the independent contributions of lymphovascular invasion (LVI) and tumor infiltrating lymphocytes (TILs) in the residual tumor bed. The methods and results are clear.

It appears form the K-M plots that there were few events in the 29 patients with luminal subtype who had pCR or RCB-I (5 RFS events and 2 OS events). It might be that there was insufficient statistical power to observe a difference in that subset - something to discuss with the statistician.

The LVI result is important and adds new information to the field. I note that a small subset had missing data and so the LVI status was imputed from other variables. It would be helpful to know whether the significance of LVI still holds when the analysis is restricted to the majority subset where LVI status was reported b the pathology review.

Some minimal edits to consider:

Page 6, Pathologic Review, 1st sentence. This sentence could be more clear. Presumably the slides from both the pre-treatment core biopsy and the post-treatment resection specimen were reviewed in all 717 cases.

Page 7, last sentence: Was this the Chi-square test?

Reviewer #3: It would be helpful to compare your findings to the findings form the Meta-analysis of > 5000 patients incl 5 and 10y EFS presented at SABCS 2019 GS5-01.

It would be helpful to include details about sampling of the breast specimens included. The SOP for RCB was published in 2007. The cohort starts in 2002. How did the sampling of the retrospectively reviewed surgical specimens differ form the RCB SOP? How did this affect calculation of the RCB scores.

6. PLOS authors have the option to publish the peer review history of their article (what does this mean?). If published, this will include your full peer review and any attached files.

Reviewer #1: No

Reviewer #2: Yes: W. Fraser Symmans, M.D.

Reviewer #3: No

---

## [Author Response · Author response to Decision Letter 0]

13 May 2020

We would like to thank the Editor for kindly accepting to review our revised manuscript. We also thank the Editor and the reviewers for the important comments and the essential revisions they suggested.

All changes were made as suggested: 

- The clinical relevance of TILS and LVI’s association with the RCB score was discussed. 

- The lack of statistical power of the pCR and RCB-I luminal BC subgroups was mentioned in the discussion, and new analyses were performed pooling together pCR and RCB-I luminal BC subgroups to address this issue (with an almost significant p-value of 0.069). 

- We also inserted a paragraph describing the meta-analysis of more than 5000 individuals RCB data with long-term follow up which was recently presented by Symmans and colleagues. 

- Analyses were reconducted only taking into account the subset of patients whose LVI status had been reported by the pathology review, with substantially unchanged results. 

- The wording of the pathology review section in materials and methods (page 6, first sentence) was modified for purposes of clarity. 

- “Khi2 tests” was replaced by “Chi-square tests” on page 7.

- The evolution of sampling methods during the time period of the cohort and its potential impact on RCB scores was discussed. No clear trend was observed when studying RCB scores before and after the publication of the RCB score in 2007. 

- In addition, we inverted the labels of the BC subtypes B and C for subtype luminal and TNBC respectively, in order to keep the same order for the BC subtype stratification throughout the manuscript.

Additional statements were added regarding :

- Financial disclosure : « The funders had no role in study design, data collection and analysis, decision to publish, or preparation of the manuscript ».

- Competing interests : « The authors have declared that no competing interests of any kind exist, with funders such as Roche France, or with other entities. This does not alter our adherence to PLOS ONE policies on sharing data and materials ». 

We thank the reviewers again for their significant contribution that substantially improved the quality of our study and the clarity of our manuscript.

---

## [Editor Report · Decision Letter 1]

21 May 2020

Prognostic value of the Residual Cancer Burden index according to breast cancer subtype: validation on a cohort of BC patients treated by neoadjuvant chemotherapy.

PONE-D-19-31211R1

Dear Dr. Reyal,

We are pleased to inform you that your manuscript has been judged scientifically suitable for publication and will be formally accepted for publication once it complies with all outstanding technical requirements.

With kind regards,

Tina Hieken, MD

Academic Editor

PLOS ONE

Additional Editor Comments (optional):

Concerns addressed and suitable for publication.
---

## [Editor Report · Acceptance letter]

29 May 2020

PONE-D-19-31211R1 

Prognostic value of the Residual Cancer Burden index according to breast cancer subtype: validation on a cohort of BC patients treated by neoadjuvant chemotherapy. 

Dear Dr. Reyal:

I am pleased to inform you that your manuscript has been deemed suitable for publication in PLOS ONE. Congratulations! Your manuscript is now with our production department. 

With kind regards,

on behalf of

Dr. Tina Hieken 

Academic Editor

PLOS ONE